# Cooperative Traffic Signal Control with Traffic Flow Prediction in Multi-Intersection

**DOI:** 10.3390/s20010137

**Published:** 2019-12-24

**Authors:** Daeho Kim, Okran Jeong

**Affiliations:** Department of Software, Gachon University, Gyeonggi 13120, Korea; ikimdh91@gc.gachon.ac.kr

**Keywords:** cooperative traffic signal control, deep reinforcement learning, traffic flow prediction

## Abstract

As traffic congestion in cities becomes serious, intelligent traffic signal control has been actively studied. Deep Q-Network (DQN), a representative deep reinforcement learning algorithm, is applied to various domains from fully-observable game environment to traffic signal control. Due to the effective performance of DQN, deep reinforcement learning has improved speeds and various DQN extensions have been introduced. However, most traffic signal control researches were performed at a single intersection, and because of the use of virtual simulators, there are limitations that do not take into account variables that affect actual traffic conditions. In this paper, we propose a cooperative traffic signal control with traffic flow prediction (TFP-CTSC) for a multi-intersection. A traffic flow prediction model predicts future traffic state and considers the variables that affect actual traffic conditions. In addition, for cooperative traffic signal control in multi-intersection, each intersection is modeled as an agent, and each agent is trained to take best action by receiving traffic states from the road environment. To deal with multi-intersection efficiently, agents share their traffic information with other adjacent intersections. In the experiment, TFP-CTSC is compared with existing traffic signal control algorithms in a 4 × 4 intersection environment. We verify our traffic flow prediction and cooperative method.

## 1. Introduction

Recently, traffic congestion has become a serious problem in most cities. Since the capacity of the road is limited, it is difficult to handle the increasing traffic flow. Traffic signal control [1,2,3] and traffic flow prediction [4,5,6] methods are the most effective solution to mitigate traffic congestion. In general, traffic signal control has the advantage of being able to cope immediately with the amount of traffic flow occurring at the intersection level [7]. On the other hand, traffic flow prediction can cover larger regions based on historical traffic data [8]. The adaptive traffic signal control [9,10] is one of the most effective ways to improve the road environment. Adaptive traffic signal control takes the action of switching the traffic phase or adjusting the traffic signal length.

Reinforcement learning is widely used in traffic signal control research to find the optimal solution [11,12]. Reinforcement learning agent learns optimal policy by exploring uncertain environment. In traffic signal control, the agent gets a state from the road such as traffic flow and takes action based on policy. The agent then receives a reward and learns to maximize the discounted cumulative reward. However, such an algorithm has a limitation that state and action space will grow exponentially as the environment size increases [13]. Due to this limitation, reinforcement learning is difficult to apply in real-world situations.

In recent years, deep reinforcement learning, which combines traditional reinforcement learning and deep neural network, was introduced. This algorithm has attracted a lot of attention because of deep Q-network (DQN) [13] applied to AlphaGo [14]. Deep reinforcement learning was first applied to game environments such as Go and Tetris, and recently to adaptive traffic signal control research. Due to high performance of DQN, many researches of traffic signal control have been implemented by DQN [15,16]. However, there is less research on deep reinforcement learning for cooperative traffic signal control at multi-intersections and a lack of possibility of applying in the real world. 

In this paper, we propose the cooperative traffic signal control with traffic flow prediction for a multi-intersection environment. The main contributions of our research are as follows:Cooperative traffic signal control method for harmonious multi-intersection environment.Traffic flow prediction for the real world.

For cooperative traffic signal control at a multi-intersection [17,18], we consider all adjacent intersections and derive a global optimal value. This method can help to derive the global optimal Q-value by estimating the probability of incoming flow from other intersections. Traffic flow prediction may predict the future traffic flow in a different way. Most existing traffic signal control studies are trained in virtual simulation [19]. In the real world, traffic flow is especially heavily influenced by variables such as weather, time, and day [20,21]. However, it is difficult to consider these variables in the virtual simulator. Thus, we propose the traffic signal control with traffic flow prediction which approaches a real environment. 

The remainder of this paper is organized as follows: The related work is presented in Section 2. The background of deep reinforcement learning algorithm and DQN extensions are presented in Section 3. The proposed cooperative methods and traffic flow prediction at multi-intersection are introduced in Section 4. Our algorithm is evaluated in various ways in Section 5. Our conclusion and future work are presented in Section 6.

## 2. Related Work

Currently, most traffic signal systems have a static traffic signal length, and static traffic signal systems do not flexibly cope with traffic changes in real-time [22]. In order to deal with these limitations, the study of adaptive traffic signal control [9,10] has been actively studied. Adaptive traffic control uses a method to switch the traffic signal phase in real-time based on the traffic volume observed at the intersection. In other words, traffic congestion can be alleviated by giving a green signal length to a lane with more traffic flow.

Various machine learning algorithms are used for adaptive traffic control [3,7,10]. Recently, many fuzzy logic-based algorithms [23,24] have been studied as machine learning algorithms. Fuzzy logic-based adaptive traffic control studies [25,26,27] generally improve the performance of existing studies. Arif A. [28] introduced an adaptive traffic signal control method using image processing techniques. This employ a method of extracting a region of interest (ROI) by recognizing cars on a road area in the image using a four-way camera installed at an intersection and performing work according to a predefined flowchart. However, the adaptive traffic signal control method also has various problems.

Traditional reinforcement learning methods are also widely used for traffic signal control. Reinforcement learning aims to find the optimal policy that maximizes the reward. In reinforcement learning, an agent explores by taking various actions based on the state and learns to maximize the reward by receiving the result of the action as a reward. YK Chin’s study [29] introduces a method for optimizing pre-defined traffic signal phases in a multi-intersection environment using the Q-learning algorithm. It describes how to optimize traffic signal control based on a total of 256 possible traffic conditions, which are categorized into four signal phases and four levels of vehicle queues, respectively, in a traffic network with two intersections. In addition, Hua W. and certain researches [16,30,31] point out the limitations of the hand-crafted rules of the current traffic signal system. They also introduced reinforcement learning, value-based deep reinforcement learning, policy gradient techniques to dynamically adjust traffic signal in response to real-time traffic congestion. 

Reinforcement learning basically selects the optimal action based on Q learning, which is not free from the “curse of dimensionality”. Therefore, deep reinforcement learning that combines reinforcement learning and an artificial neural network is proposed [13].

Deep reinforcement learning has received a lot of attention in solving problems such as the “curse of dimensionality”. DQN showed effective performance by approximating high action space through the neural network. Due to the high performance of DQN, deep reinforcement learning has been applied to adaptive traffic signal control [32,33]. It is especially possible to utilize all the state information of intersection through CNN [32]. Since DQN, various extensions such as Double DQN [34] and Dueling DQN [35] have been introduced quickly, and the research [36] combining these extensions improved the learning stability much more than the existing DQN.

Recently, multiple intersection traffic signal control has been studied [18,19,20,21,22,23,24,25,26,27,28,29,30,31,32,33,34,35,36,37], but there is less research of traffic signal control at multiple intersections. Ref [16] proposes an A3C technique with a decentralized coordinated algorithm for multi-intersection traffic signal control. CNN is used to extract vehicle position and velocity information from a snapshot of the simulator. However, most snapshots are inefficient because they do not include the vehicle, resulting in sparse matrix. Ref [37] applied DQN as a traffic signal control algorithm and proposes Q-value transfer for harmonious multiple intersections control. However, several DQN extensions have already appeared that improve on the shortcomings of DQN, and in many areas have been proven to outperform existing DQN. In addition, because the experimental environment is less than 10 intersections, it is difficult to apply in real-world situations. In addition, the existing traffic signal control studies were conducted using a virtual simulator, which does not deal with variables such as temperature and weather that actually affect traffic conditions.

In this paper, we propose an algorithm that combines cooperative traffic signal control at multiple intersections and further predicts traffic volume taking into account real-world variables. For cooperative traffic control at multiple intersections, we consider traffic information at adjacent intersections. We estimate the global optimal q-value at multiple intersections by sharing the local optimal q-values of each adjacent intersection. We also consider the various variables that affect actual traffic, thus enabling more advanced training. In an experiment, we demonstrate high performance by applying algorithms combined with various DQN extensions to traffic signal control.

## 3. Deep Reinforcement Learning and Extensions to DQN

In reinforcement learning, the agent learns an optimal policy by interacting with the environment. The learning process of reinforcement learning is illustrated in Figure 1. At each step, the agent receives the state from an uncertain environment. Then the agent selects the action based on policy. The agent then receives a reward from the environment. The goal is to maximize the cumulative reward. The Q-function predicts the expected future reward given state and action. The Q-function is formulated as follows:(1)$$Q^{\pi }\left(s,a\right)=E\;\left[\sum_{k=0}^{\infty }\gamma^{k}r_{t+k}\;|\;s_{t}=s,\;a_{t}=a,\;\pi \right]$$
where γ is a discount factor and is to multiply the expected reward, it means that set the lower reward value received at later.

### 3.1. Deep Reinforcement Learning

In reinforcement learning, it is important that how to define state, action, and reward function, respectively. The state, action, and reward are defined as follows.

#### 3.1.1. State

The state represents the observation information from environment. Since the DQN uses the deep neural network, which is a non-linear approximator, many researchers define snapshot of simulator as state [32,33]. The snapshot of simulator is useful to get the vehicle status such as speed and position of vehicles. However, it needs additional state encoding process to get the traffic state space. In this paper, we simply set the number of vehicles of each lane as state space. Figure 2 shows the example of an intersection. Each intersection has four-arms (edges) and each edge has two lanes. To deal with multi-intersection scenario, we must get the number of vehicles in each lane at each time step.

#### 3.1.2. Action

At each time step, the agent selects the action from given state. In this paper, we define a set of actions A, A = {NSG, EWG, NSLG, EWLG}. Each action of set represents the traffic signal phase and N, S, E, W, L, G represents North, South, East, West, Left, Green and where NSG indicates North-South Green, EWG indicates East-West Green, NSLG indicates North-South Left Green and EWLG indicates East-West Left Green. The example of a traffic signal system is illustrated in Table 1 and the traffic signal system of all intersections are the same. Table 1 shows traffic signal phases for each action {NSG, EWG, NSLG, EWLG}. There are 8 phases and each phase consists of traffic signal such as G(Green), y(yellow) and r(red) in the order R(Right turn lane), S(Straight lane) and L(Left turn lane). For example, phase 0 means that turn on the green signal to north and south, straight and right lanes. Also, each traffic signal phase has a duration. In this paper, we select the traffic signal phase based on observed state. For example, in Figure 2, the number of vehicles in the north and south is greater than in east and west. Thus, if the NSG (= phase 0) phase is already set, keep that phase and increase the duration, otherwise switch to NSG phase and increase the duration. In addition, in order to consider inherent influence between intersections, we propose a cooperative method that shares traffic conditions between intersections. Figure 5 shows an example of our cooperative method. If the traffic signal at intersection 3 is EWG (East-West Green) and the traffic signal at intersection 8 is NSG (North-South Green), intersection 4 is naturally affected by the traffic coming from intersections 3 and 8. To take this effect into account, we propose a cooperative method of transferring the state and action of adjacent intersections to other intersections.

#### 3.1.3. Reward

The reward function is the most important element that differentiates other machine learning algorithms. The reward function is used to evaluate the model performance. Since the goal of reinforcement learning is to maximize the reward, defining reward function is important. In this paper, we define reward function as the changes of average waiting time between time step *t* and time step *t*+1. The lower the waiting time, the higher reward. The reward function r is formulated as follows:(2)$$R_{waiting\_time}=W_{t}-\;W_{t+1}$$

### 3.2. Extensions to DQN

Deep reinforcement learning has achieved remarkable success in many domains from game environment to traffic signal control. DQN [13] is representative deep reinforcement learning algorithm and shows the high performance. Due to the encouraging performance of DQN, various extensions of DQN [34,35] have been introduced. Recently, [36] proposed the combination of extensions and demonstrate high performance in the Atari game environment. They combine 6 extensions with DQN which are Double Q-learning, Dueling network, Prioritized replay [38], Multi-step learning [39], Distributional RL [40] and Noisy Nets [41]. We applied that those algorithms to cooperative traffic signal control and we explain how they are applied to our algorithm. 

Double Q-learning improved the DQN by decoupling the selection of the action from its evaluation. This algorithm improved an overestimation problem of conventional Q-learning. In our model, we construct the separate target network from main network. Dueling network is a neural network architecture designed for value-based reinforcement learning. This algorithm divides neural network into value and advantage stream. That is, the Q function of DQN is divided into state value functions and advantage function. This algorithm can get faster learning speed because the advantage function only focuses on the action value. We applied dueling network when constructing neural network in our model. Prioritized replay improves data efficiency, by replaying more often transitions from which there is more to learn. The priority of transition is calculated based on TD (Temporal Difference) error. At each training step, we calculate the priorities of transitions and sampled the high priority transitions. Multi-step learning truncates the n-step rewards, not the single reward. Instead of computing the reward function at every training step, we compute the reward n-steps after. This algorithm is able to minimize the loss and lead to faster learning. Distributional RL is also related to reward. DQN and other algorithms commonly define average waiting time or delay as reward. Thus, reward value is scalar. Instead of a scalar reward, we learn to approximate the distribution of the reward. This algorithm is useful when the future reward is complex and has multimodal features by representing the reward in distribution. NoisyNet proposes a noisy linear later that combines a deterministic and noisy stream. Over time, the network can learn to ignore the noisy stream, but will do so at different rates in different parts of the state space, allowing state conditional exploration with a form of self-annealing. NoisyNet is useful when action space is too large and replaces the existing exploring policy, є-greedy. Algorithm 1 shows a description of the whole algorithm and how extensions of DQN such as Double DQN, Dueling DQN, Prioritized experience replay, Distributional RL, Multi-step learning, and NoisyNet were harmoniously applied to traffic signal control.
**Algorithm 1.** Cooperative traffic signal control.Initialize network parameter θInitialize target network parameter θ−Initialize replay memory *m* with size *M*, batch size B, target network replacement frequency *T*Initialize initial state *s*for epochs *e*
∈ {1, … *N*} do Observe current state st Select an action *a* from noisy network Get reward *r* and next state st+1 If |*m*| > *M* then  Delete oldest transition *t* from *m* end Add transition *t* = (st, *a, r*, st+1) to replay memory *m* st ← st+1 If |*m*| > *B* then  Sample *B* transitions from m based on prioritized experience replay  Compute multi-step loss based on distributional loss  Update network θ using loss  If *e* == *T* then   Update target network θ− ← θ  end endend

## 4. TFP-CTSC (Cooperative Traffic Signal Control with Traffic Flow Prediction)

In this section, we propose the cooperative traffic signal control with traffic flow prediction (TFP-CTSC) in a multi-intersection. There are two reasons why we propose our paper: (1) There is less research on traffic signal control in multi-intersections. (2) Existing studies of traffic signal control are dependent on virtual simulation environment. To efficiently improve the performance in multi-intersection environment, we consider the adjacent intersections at the last time step. Also, we complement the limitations of virtual simulation by considering realistic variables that affect traffic flow. Overall architecture of our paper is illustrated in Figure 3. At each time stept, the agent receives the state from environment. At the same time, agent send state to traffic flow prediction model. In traffic flow prediction model, they predict the future traffic flow (state_t+1_) using realistic variable. And then, return the state_t+1_ to agent. Agent compute the optimal Q-value from both observed state and predicted future traffic flow to choose the action. For a state and action given, the reward is received. The main goal of our model is also to find the optimal policy which maximize the future reward.

### 4.1. Traffic Flow Prediction for Real-World

In the real world, traffic flow can be easily affected by those variables such as weather, day, and time. For example, generally, traffic congestion usually occurs more often at rush hour and in weather such as snow or rain [21]. In a simulation environment, it is difficult to deal with these factors. That’s why we construct separate traffic flow prediction model to reflect real-world variables. We select the day, time, and temperature that most influence the traffic flow, and the correlation matrix between each variable is shown in Figure 4a. We can see that highly correlated variable with t+1-flow are t-flow, temperature, and time.

Traffic flow prediction research has same objective as traffic signal control to mitigate the traffic congestion by predicting the future traffic. Our model is trained by using LSTM (Long-Short Term memory) [42] to predict. LSTM is useful to training the time-series data because of a memory layer.

When the state _t_ is received from environment, traffic flow prediction model predicts the state _*t*+1_ at time step *t*+1. The predicted state *_t_*_+1_ is used to optimize the Q-value. We use RMSE (Root Mean Squared Error) as loss function. The loss is 12.9 and RMSE is 0.023. The Q-values are updated as follows:(3)$$s\prime_{t+1}=f_{t}\left(time,\;day,\;temperature,\;S_{t}\right)\;$$
(4)$$a^{\prime }=argmaxQ_{t}^{i}\left(s\prime_{t+1},a\right)$$
(5)$$Q_{t+1}^{i}(s_{t}^{i},a_{t}^{i})=\;Q_{t}^{i}(s_{t}^{i},a_{t}^{i};\theta_{i})+\eta *(R_{t+1}+\;\gamma maxQ_{t}^{i}\left(s\prime_{t+1},a^{\prime };\theta \prime_{i}\right)-Q_{t}^{i}(s_{t}^{i},a_{t}^{i};\theta_{i}))$$
here $s_{t+1}\prime$ is predicted traffic flow of traffic flow prediction model at time step *t*, $a^{\prime }$ is indices of maximum value in $Q_{t}^{i}\left(s\prime_{t+1},a\right)$, $\theta_{i}$ and $\theta \prime_{i}$ are the parameters of evaluation network and target network, respectively. Our updated Q-value replaces $s_{t+1}$ with predicted traffic value of traffic flow prediction model. This newly updated Q-value is getting closer to real-world by traffic flow prediction. The Figure 4b shows the visualized results of traffic flow prediction. The predicted traffic flow is represented as a circle. The closer the circle is to red, the higher traffic flow and the closer to green, the lower traffic flow.

### 4.2. TFP-CTSC in Multi-Intersection

For cooperative traffic signal control in multi-intersection, we propose transfer planning [43] based cooperative traffic signal control with traffic flow prediction. We model a multi-intersection environment as a multi-agent. Each agent is trained to find local optimal Q-value, and then we try to find global optimal Q-value by transferring the messages which contain local optimal value of each agent. Cooperative traffic signal control is possible by considering not only its own Q-value, but also the Q-value of adjacent intersections. Finally, we can draw the global optimal Q-value of each agent and this Q-value is updated as follows:(6)$$Q_{t+1}^{i}(s_{t}^{i},a_{t}^{i})=\;Q_{t}^{i}(s_{t}^{i},a_{t}^{i};\theta_{i})+\eta *(R_{t+1}+\;\gamma maxQ_{t}^{i}\left({s^{\prime }}_{t+1},a^{\prime };{\theta^{\prime }}_{i}\right)-Q_{t}^{i}(s_{t}^{i},a_{t}^{i};\theta_{i}))\;+\;\underset{j\in N_{adj}}{\sum }Q_{t-1}^{i}\left(s_{t-1}^{j},a_{t-1}^{j};\theta_{j}\right)$$
where $N_{adj}\;$ is the number of adjacent intersections and $\theta_{j}$ is parameter of evaluation network of intersection j.

## 5. Experiment

### 5.1. Experimental Setting

To evaluate the performance of cooperative traffic signal control with traffic flow prediction (TFP-CTSC) in a multi-intersection, the experiments are conducted on 4 × 4 road network scenarios as shown in Figure 5. Figure 5 shows our experiment road network and 2 × 2 intersection example which is part of a 4 × 4 intersection. TFP-CTSC is experimented in 16-intersection environment and each intersection is modeled as reinforcement learning agent. For cooperative traffic signal control, the number of vehicles in adjacent intersection are considered by transferring the state. We use Simulation of Urban Mobility (SUMO) [44] which is widely used in traffic signal control research. To receive the traffic status information, we use Traci API [45] supported in SUMO. TFP-CTSC is implemented using Pytorch which is python deep learning library. The hyperparameters for TFP-CTSC and configurations of road scenario and SUMO are shown in Table 2 and Table 3, respectively. In Table 2, from Batch size to Epoch are hyperparameters used for existing DQN, Alpha to prioritized epsilon for prioritized experience replay, Vmin to Atom size for distributional RL, and Multi-step size for multi-step learning. Table 3 shows the configurations of the road simulation environment built with SUMO.

### 5.2. Experimental Methodology

TFP-CTSC is trained to maximize the rewards. We define waiting time as a reward described before. In this experiment, we experimented with queue length and delay in addition to the waiting time used as a reward function to prove how effective the proposed algorithm is for improving overall traffic performance. The overall experimental structure and procedure is shown in Figure 6. The SUMO simulator creates a road environment using configuration files. In the road environment, vehicles are generated and observed periodically, and the observed vehicles are delivered to agents which were assigned to each intersection. Each agent selects the best traffic signal phase and switches the phase based on observed traffic volume. Through road conditions after traffic signals are switched, parameters such as the number of vehicles, queue length, waiting time, and delay are observed. The lower the waiting time, queue length, and delay, the better the performance. In the first experiment, we compare TFP-CTSC with existing reinforcement learning algorithms which are DQN, Q-learning. Through the first experiment, how well we define the reward function and our algorithm shows high performance in each three performance metrics. In the second experiment, we evaluate our main contributions which are cooperative method in multi-intersection and traffic flow prediction. We compare TFP-CTSC with the model, ignoring the traffic flow prediction and cooperative methods. In the final experiment, we evaluate the model performance according to the definition of the reward function most important in reinforcement learning.

### 5.3. Experimental Results

#### 5.3.1. Comparative Analysis with Existing Algorithms

To evaluate TFP-CTSC, we compare our model with an existing reinforcement learning algorithm including DQN and Q-learning algorithm. The lower waiting time, queue and delay, the better the performance. Figure 7, Figure 8 and Figure 9 shows the overall performance of traffic signal control in a 4 × 4 intersection scenario for different performance metrics. It can be seen that TFP-CTSC model outperforms the other model. In three experimental results, we can see waiting time, queue length, and delay of TFP-CTSC is lower than other algorithm. And TFP-CTSC does not have much fluctuation compared to other models. That is, as the learning progresses, TFP-CTSC is more stable in both results of waiting time and queue. 

#### 5.3.2. Comparative Analysis for the Effect of TFP and CTSC

Our main contributions are cooperative traffic signal control (CTSC) and traffic flow prediction (TFP). To evaluate the effectiveness of TFP-CTSC, we compare TFP-CTSC with a model without CTSC and without CTSC and TFP. Figure 10a shows the result of comparing the average waiting time between each model and Figure 10b shows the result of queue length. In both Figure 10a,b, the TFP-CTSC outperformed the other models. Overall, TFP-CTSC shows lower waiting time and queue length, and tends to decrease as the training progresses.

#### 5.3.3. Comparative Analysis of Reward Function

In this experiment, we compare the performance of the model according to how we define the reward function. The object of comparison is the reward function which considers only waiting time and the reward function, which considers both waiting time and queue length. Figure 11 shows the average reward at each intersection of two reward functions. The reward function considering only waiting time showed a high average reward at all intersections. This means that the reward function we defined has high performance and how to design the reward function is important.

## 6. Conclusions

In this paper, we propose a cooperative traffic signal control combined with traffic flow prediction (TFP-CTSC) at multiple intersections. Our traffic signal control algorithm has various extensions of DQN (Deep Q-Network) for fast learning speed and stable learning. We also introduce a method for cooperative traffic signal control to address the limitation of the lack of traffic signal control studies in a multi-intersection. Each intersection is modeled as one agent for cooperative traffic signal control, and each agent estimates the local Q-value. By transferring local Q-values between intersections, we can estimate global optimal Q-values. In addition, in order to consider variables that affect the actual traffic status, a traffic flow prediction model was separately constructed. The future traffic flow predicted by traffic flow prediction supports finding an optimal policy that maximizes the reward of the reinforcement learning model. To evaluate our model, we conducted various parametric experiments with existing reinforcement learning algorithms and demonstrated high performance. Also, to verify the effectiveness of our main contributions, cooperative method and traffic flow prediction, the proposed model is compared with the model without cooperative method and traffic flow prediction. As a result, this paper proposes cooperative traffic signal control combined with traffic flow prediction to mitigate traffic congestion at multiple intersections. In future work, we will study algorithms that cover wider road networks such as the city and improve them for practical applications.

## Figures and Tables

**Figure 1 sensors-20-00137-f001:**
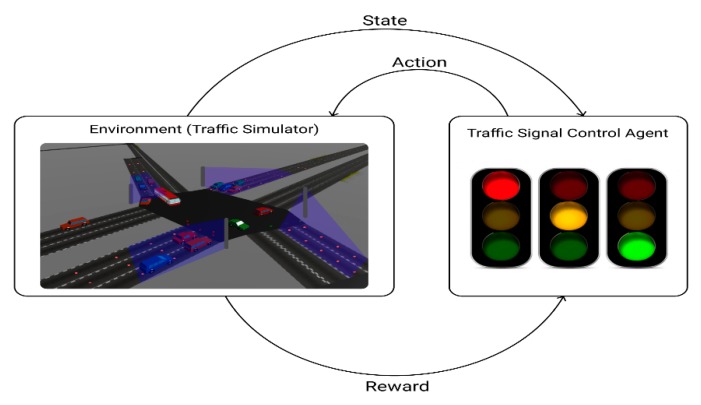
Reinforcement learning based traffic signal control.

**Figure 2 sensors-20-00137-f002:**
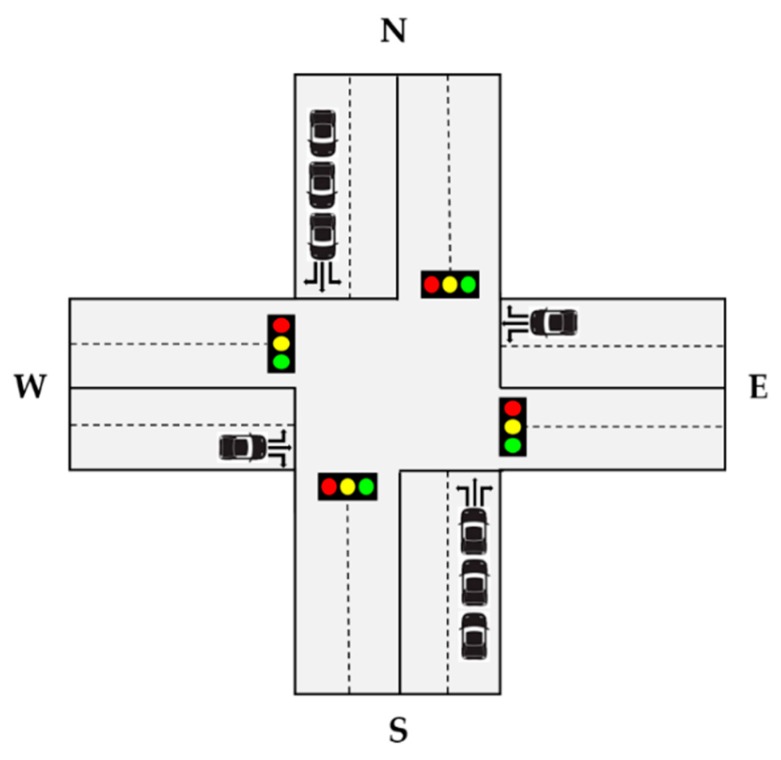
The 4-arm structure of intersection.

**Figure 3 sensors-20-00137-f003:**
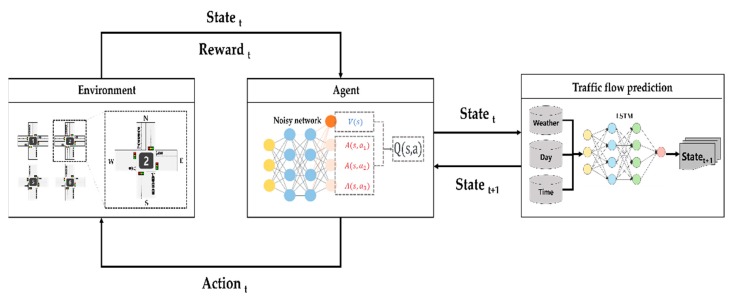
Overall architecture of TFP-CTSC.

**Figure 4 sensors-20-00137-f004:**
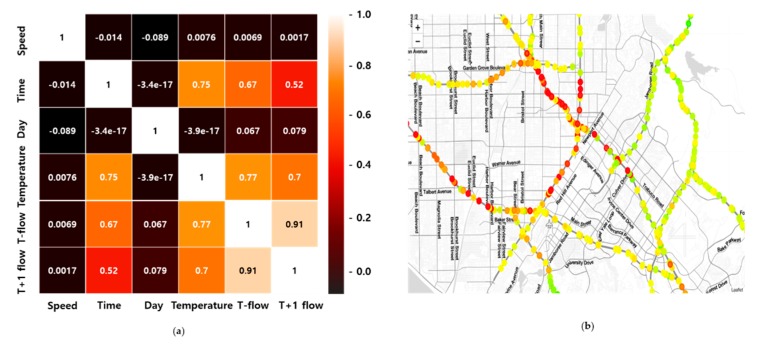
(**a**) Correlation matrix and (**b**) Visualized results of traffic flow prediction.

**Figure 5 sensors-20-00137-f005:**
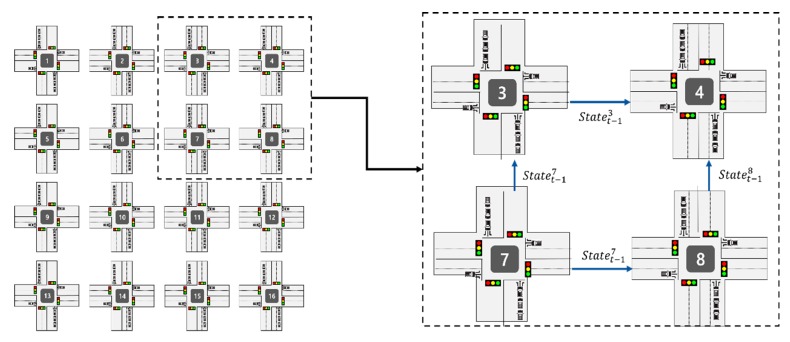
4 × 4 intersection road network.

**Figure 6 sensors-20-00137-f006:**
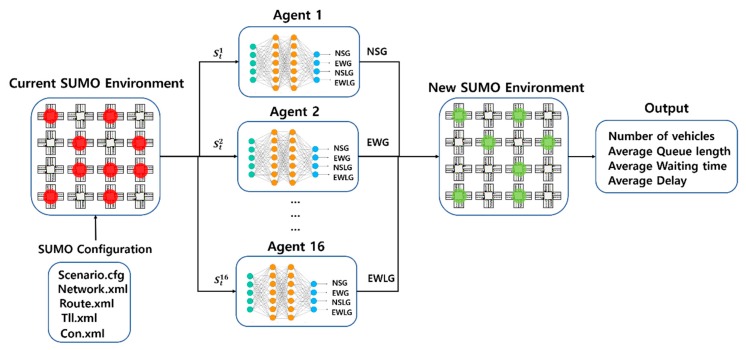
Experimental structure and procedure.

**Figure 7 sensors-20-00137-f007:**
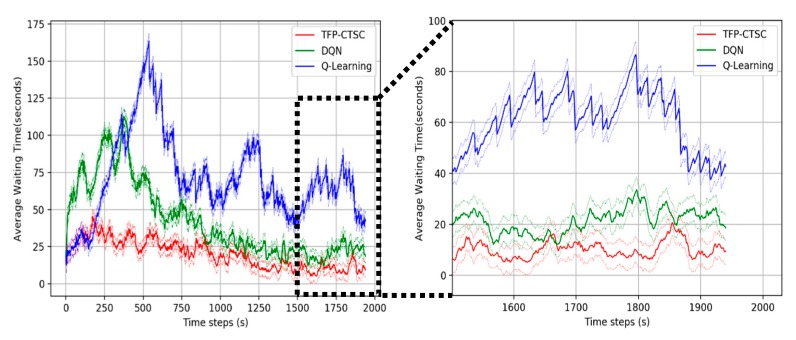
Performance comparison for average waiting time.

**Figure 8 sensors-20-00137-f008:**
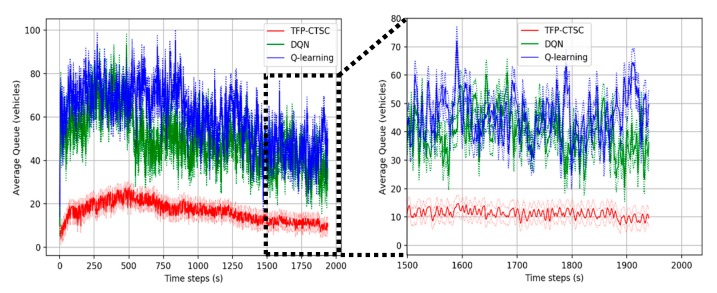
Performance comparison for queue length.

**Figure 9 sensors-20-00137-f009:**
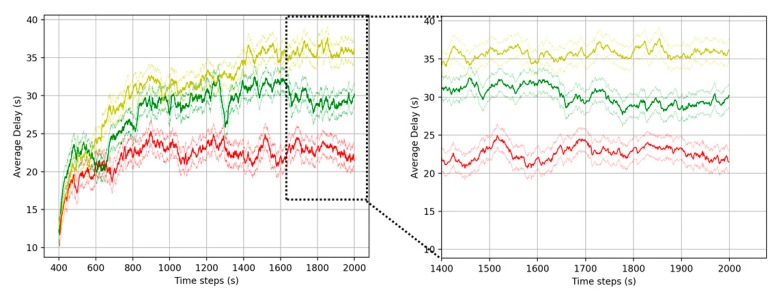
Performance comparison for delay.

**Figure 10 sensors-20-00137-f010:**
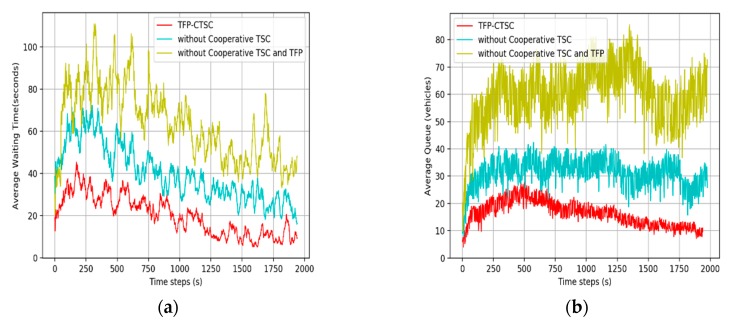
Performance comparison with TFP-CTSC without CTSC and TFP. (**a**) Average waiting time; (**b**) queue length.

**Figure 11 sensors-20-00137-f011:**
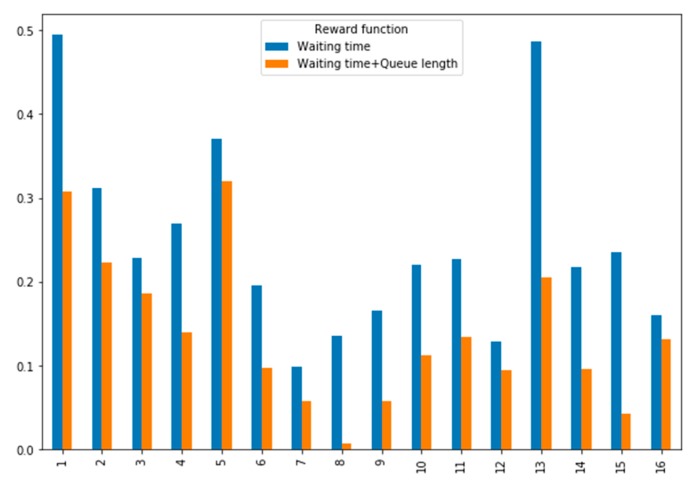
Performance comparison for reward function.

**Table 1 sensors-20-00137-t001:** The example of traffic signal system at single intersection.

Traffic Signal ID = ‘2’													
	Dir	N			E			S			W		
Phase		R	S	L	R	S	L	R	S	L	R	S	L
Phase 0 **(NSG)**		G	G	r	r	r	r	G	G	r	r	r	r
Phase 1		y	y	y	r	r	r	y	y	y	r	r	r
Phase 2 **(NSLG)**		y	y	G	r	r	r	y	y	G	r	r	r
Phase 3		r	r	y	r	r	r	r	r	y	r	r	r
Phase 4 **(EWG)**		r	r	r	G	G	r	r	r	r	G	G	r
Phase 5		r	r	r	y	y	r	r	r	r	y	y	r
Phase 6 **(EWLG)**		r	r	r	r	r	G	r	r	r	r	r	G
Phase 7		r	r	r	r	y	r	r	r	r	r	y	r

**Table 2 sensors-20-00137-t002:** Hyperparameters for TFP-CTSC.

Hyperparameter	Description	Value
Batch size	Size of batch for sampling	32
Learning rate	Step size in loss function	0.0001
Discount factor	Weight that multiplies future reward	0.99
Replay memory size	Size of Replay memory	1000
Optimizer	Optimization algorithm	Adam
Target network replacement frequency	How frequently update target network	100
Epoch	Training step size	2000
Alpha	How much prioritization used	0.2
Beta	How much importance sampling is used	0.6
Prioritized epsilon	How much every transition can be sampled	1e-6
Vmin	Minimum value of support	-10
Vmax	Maximum value of support	10
Atom size	The unit number of support	51
Multi-step size	Step size to calculate n-step reward	3

**Table 3 sensors-20-00137-t003:** Configuration of Road scenario and SUMO.

Hyperparameter	Value
Intersection size	16
Number of lane	2
Simulation start	0 s
Simulation end	35,999 s
Vehicle speed	13.00 km/h
Duration of yellow	2.0 s
Minimum duration of green	3.0 s
Maximum duration of green	50.0 s

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
