# Peer review of "Cooperative Traffic Signal Control with Traffic Flow Prediction in Multi-Intersection"

_sensors, 2019, doi:10.3390/s20010137_

Round 1

Reviewer 1 Report

I believe there is a main big problem with this paper. In the title of the paper and in the text it is claimed that the control strategy applied is :"Cooperative traffic signal control". The cooperation in the traffic control system proposed is between different intersections.

Traditionally the term "Cooperative traffic control" is currently used to indicate control strategies applied on the base of "connected" vehicles. Connected vehicles can communicate with infrastructure (Vehicle to Infrastructure) and among them (Vehicle to Vehicle).

In other words the cooperation is between the vehicles!!!!

There is no such kind of vehicles in the methodology proposed. The cooperation that is presented should be simply defined "Coordination". I believe the term cooperative has been misused in this paper. There is nothing new in coordinating intersection signals.

The main novelty of the paper is only the application of Deep Q-Network.

The paper is interesting and should be rewritten taking into consideration the main point above and other points:

1-Reference should be given to papers that discuss cooperative (competitive) signal traffic control and the authors have to clarify the differences between their method with other works that are based on a "cooperative" "competitive" paradigm.

2-The experimental setting description is too small. Is not clear how it was done and the way SUMO has been used. How was the microsimulation model calibrated?

3-A diagram showing the inputs and outputs of SUMO and the inputs and outputs of the control procedure would be helpful.

Reviewer 2 Report

In this paper, authors proposed a cooperative method to solve the traffic congestion problem via traffic lights signal control for harmonious multiple intersections. The method mainly considers the traffic flow prediction results’ influence on the state equation in DQN. The architecture and the basic idea of the method is reasonable. However, there are some drawbacks that need to be improved:

Regarding the definitions of ACTION in 3.1.2, please explain the reason that parts of actions have 3 elements, e.g. NSG, the other parts have 4 elements, e.g., NSLG. Please add the definitions of ‘y’ in the table 1. How to deal with the complexity problem as the number of intersection’s increasing? As shown as table1, in only 2*2 intersection scenario, the number of elements in this matrix has become 8*4*3. The volume will become huge in the real-world road networks that includes complex intersections. In the reward function of 3.1.3, the authors considered two parameters, the length of waiting queue and the waiting time. However, the reward function cannot sum simply length and time. Maybe some improvements for the reward function are necessary to guarantee the measurement units of the two parameters are the same. Comparing to the intelligent path planning to the drivers, what is the advantage of controlling the traffic lights’ signal? And how to deal with the problem that the most drivers choose the segments/intersections that has longer green lights? The authors should emphasize the motivation that using the deep reinforcement learning method to solve this problem. Moreover, the current revision lacks the descriptions of the whole algorithm. Please add some explanations or pseudo code for the processes of the proposed algorithm. The evaluations are weak, which only considered the waiting queue and waiting time as the metrics. They are just the parameters of the reward function. If the final ambition of this method is solving the traffic congestion problem, the evaluation should aiming to discussion the traffic performance based on the proposed algorithm.

Overall, we recommend a major revision for this paper.

Round 2

Reviewer 2 Report

the authors improved the manuscript in both model and simulation.  Most of the problems have been solved.  However, two of them needs authors a final check.   Regarding commen3, if the table 1 represents only one intersection's conditions, author should also consider the inherent influence among different intersections.   Regarding commen4, the reward function is not  improved in the revised version.  It just simplied.   overall, we recommend author consider and check the model more carefully.  It is better to do some minor improvements before acceptance.
